# Snacking on Whole Almonds for Six Weeks Increases Heart Rate Variability during Mental Stress in Healthy Adults: A Randomized Controlled Trial

**DOI:** 10.3390/nu12061828

**Published:** 2020-06-19

**Authors:** Vita Dikariyanto, Leanne Smith, Philip J. Chowienczyk, Sarah E. Berry, Wendy L. Hall

**Affiliations:** 1Department of Nutritional Sciences, School of Life Course Sciences, Faculty of Life Sciences and Medicine, King’s College London, Franklin-Wilkins Building, 150 Stamford Street, London SE1 9NH, UK; vita.dikariyanto@kcl.ac.uk (V.D.); leanne.smith@kcl.ac.uk (L.S.); sarah.e.berry@kcl.ac.uk (S.E.B.); 2Department of Clinical Pharmacology, School of Cardiovascular Medicine and Sciences, Faculty of Life Sciences and Medicine, King’s College London, St Thomas’ Hospital, London SE1 7EH, UK; phil.chowienczyk@kcl.ac.uk

**Keywords:** almonds, randomized controlled trial, heart rate variability, cardiovascular disease, snacking, stress

## Abstract

Cardiac autonomic regulation can be indirectly measured by heart rate variability (HRV). Low HRV, which can be induced by mental stress, is a predictor of risk of sudden cardiac death. Few studies have investigated cause-and-effect relationships between diet and HRV. Nut consumption is associated with CVD risk reduction, but the impact on HRV, particularly in response to stress, is unclear. Men and women (30–70 y) with above average risk of developing CVD were randomly assigned in a 6-week randomized, controlled, parallel arm trial to consume either whole almond or isocaloric control snacks (20% of daily estimated energy requirement). Control snacks contained the average nutrient profile of UK snacks. Five-minute periods of supine heart rate (HR) and HRV were measured at resting and during mental stress (Stroop color-word test) at baseline and six weeks. High frequency (HF) power, which reflects parasympathetic regulation of HR, was increased following almonds during the mental stress task relative to control (mean difference between groups 124 ms2; 95% CI 11, 237; *p* = 0.031, *n* = 105), but other indices were unaffected. Snacking on whole almonds instead of typical snacks may reduce risk of CVD partly by ameliorating the suppression of HRV during periods of mental stress.

## 1. Introduction

Mental stress, increasingly a feature of modern fast-paced lifestyles, is recognized as contributing to the risk of developing cardiovascular disease [1,2]. Mental stressors stimulate cardiovascular responses which can be indirectly assessed via heart rate variability (HRV), a measure of fluctuation in the length of interbeat interval (IBI), modulated by the dynamic regulation of autonomic nervous system (ANS). The ANS consists of two branches that control variations in heart rate (HR), the sympathetic nervous system (SNS) which is activated by conditions of stress and increases HR, and the parasympathetic nervous system (PNS), specifically the vagus nerve, which counteracts the SNS and is dominant under conditions of rest, decreasing HR. Mental stress also attenuates baroreflex sensitivity [3] and increases endothelial dysfunction [4], which may be followed by slower recovery of HRV leading to the development of CVD risk [5,6]. Evidence from neuroimaging studies shows that HRV is associated with parts of the brain that regulate emotional responses [7], and greater cardiovascular reactivity in response to mental stress and slower recovery from stress are longitudinally linked to impaired cardiovascular risk status, including higher BP, hypertension and greater carotid intima-media thickness (subclinical atherosclerosis) [8].

HRV can be reported using the analysis of time- and frequency-domain parameters [9], as well as non-linear parameters, which are able to assess complexity of patterns and randomness in the HRV signal. Low HRV indicates a dominance of sympathetic activity and suppressed parasympathetic control; prolonged low HRV indicates impaired regulation of HR in response to dynamic demands and is associated with increased risk of cardiovascular morbidity and mortality in both the general healthy population [10,11] and patients with coronary heart disease (CHD) [12,13,14,15]. Conversely, high HRV suggests resilience of the cardiac ANS in adapting to stress and is inversely associated with cardiovascular disease (CVD) [16].

Lifestyle factors, such as diet and levels of physical activity, may influence HRV [17]. Scientific understanding of the impact and mechanism of dietary modifications on HRV is still very limited [17], and mainly relates to the effects of omega-3 supplementation [18,19]. During times of stress, people are more inclined to seek reward in the shape of snack foods with a less favorable nutrient profile, for example, higher in saturated fats, refined starch, free sugars, and sodium [20,21]. Substituting typical snacks with healthier snacks may ameliorate the reduction in HRV following a stress stimulus by both these pathways. Improvements in diet quality may enhance HRV both by improving neurological function and by directly influencing the responsivity of the target organ (e.g., the heart) to parasympathetic efferent input. Diet may also indirectly enhance HRV through improving mental health and reducing psychological stress, thereby reducing excess sympathetic activity [17].

In this study, almonds are utilized as a model healthy snack, since they have previously been shown to have cardioprotective effects using intermediary risk factors. Almonds, the most consumed tree nut globally [22], mostly as a snack food, are rich in unsaturated fats, dietary fiber, potassium, magnesium and vitamin E, and low in saturated fatty acids, sugar and sodium [23]. Almond consumption can improve intermediary cardiovascular risk factors [24,25,26], such as low-density lipoprotein (LDL), apolipoprotein B (apo-B), blood pressure and adiposity [27]. Using a randomized, controlled, parallel 6-week arm study design [28], we aimed to investigate the impact of replacing usual snacks with whole almonds on the HRV response to stress in a healthy, free-living adult population sample who were at above average risk of cardiovascular disease. It was hypothesized that snacking on almonds, relative to control snacks, would result in higher HRV during a laboratory mental stress test.

## 2. Materials and Methods

### 2.1. Study Subjects

The ATTIS (Almond Trial Targeting dietary Intervention with Snacks) study was specifically designed to include male and female adults aged 30–70 y with moderate risk of developing CVD that were self-reported regular snack consumers (consumed 2 or more snacks per day) around London. Identification of the risk of developing CVD was conducted following the Framingham risk score system, based on total plasma cholesterol (TC), high-density lipoprotein (HDL) cholesterol, blood pressure, and body mass index (BMI)/waist circumference (WC); moderate risk was defined as having score 2 or more [29].

Exclusion criteria included self-reported prediabetic or diabetic condition; history of heart attack (myocardial infarction) or stroke, cancer (excluding basal cell carcinoma) in the last five years, epilepsy or regular fainting, cholestatic liver disease, pancreatitis, alcohol or drug abuse; diagnosis of cardiovascular problems, angina, thrombosis, pacemaker, gastrointestinal disorders, renal or bowel disease; use of a drug likely to alter gastrointestinal motility or nutrient absorption; presence of metal inside the body; allergy or intolerance to nuts; currently pregnant, planning pregnancy, breastfeeding or having given birth in the preceding 12 months; unwillingness to follow the protocol and/or give informed consent; weight change of >3 kg in preceding 2 months; body mass index (BMI) of <18 kg/m^2^ or >40 kg/m^2^; current smokers or individuals who quit smoking within the last 6 months; and participation in other research trials involving dietary or drug intervention and/or blood collection in the past 3 months. 

Participants were recruited from the general population in the London area. Eligible potential participants were invited to a screening visit at King’s College London (KCL). Anthropometry (body height and weight, WC, body fat composition by bioelectrical impedance) and clinic blood pressure (cBP) were measured, and fasted blood samples were collected for analysis of blood glucose, full lipids, liver function and full blood count.

### 2.2. Study Design

The ATTIS study [28] was approved by the UK National Research Ethics Service (REC 16/LO/1910, approved 24 January 2017), registered with ClinicalTrials.gov (NCT02907684), and run in accordance with the Declaration of Helsinki 1975, revised in 2013, and the principles of Good Clinical Practice. Data collection was carried out between June 2017 and May 2019. All subjects gave their informed consent for inclusion before they participated in the study. The study was primarily designed to investigate the effects of almond snack consumption on endothelial function and liver fat, with HRV as a secondary outcome.

The trial used a randomized, controlled, parallel design in a free-living cohort. Participants were randomly assigned to one of two intervention arms, almonds or control. The research coordinator conducted treatment allocation using minimization software (MinimPy 0.3, Copyright (c) 2011 Mahmoud Saghaei, http://minimpy.sourceforge.net/), with age, sex, ethnicity, cardiometabolic score and willingness to undergo MRI/MRS scan as minimization variables (base probability 0.7, 1:1 allocation ratio). A 2-week run-in period consuming control snacks preceded randomization to ensure study subjects were able to tolerate the study protocol prior to starting the intervention period and to collect 4-day food diary and physical activity data. Following run-in, participants attended baseline visits at the Clinical Research Facility, St Thomas’ Hospital, London, for measurements of HRV in the resting state and during a mental stress test, as well as providing a fasted blood sample, having anthropometric measurements, and other procedures to assess changes in cardiometabolic risk factors. At 2 and 4 weeks of intervention, participants attended the Metabolic Research Unit at KCL for measurements of body composition and they also completed 24 h dietary recalls to monitor compliance. Following 6 weeks of dietary intervention, participants attended the endpoint visit at the Clinical Research Facility, St Thomas’ Hospital, and underwent the same measurements and procedures as at the baseline visit. A further 4-day food diary was completed. Before baseline and endpoint clinic visits, subjects were requested to avoid alcohol and strenuous activity for 24 h, and to consume low-fat meals and no alcohol in the evening before. Due to the nature of the intervention, participants and research staff administering interventions and conducting physiological and anthropometric measurements on study days were not blinded, but blood biomarker analysis and statistical analysis were conducted by analysts blinded to treatment allocation.

### 2.3. Intervention

Snacks usually contribute to 20–25% of energy intake [20,30,31]. Thus, the dietary intervention included a 2-week run-in period, wherein participants consumed control snacks equaling 20% of estimated energy requirements (EER), followed by random allocation to either control or almond snacks at 20% of EER for 6 weeks. EER was calculated using Henry equations and physical activity level (PAL) was estimated from 4-d activity diaries [32]. Control snacks were sweet and savory mini-muffins, baked at KCL, and were formulated and developed to be representative of the average macronutrient intakes from snacks (excluding fruit) in the UK National Diet and Nutrition Survey population [20], as detailed elsewhere [28]. Almond snacks were dry-roasted whole non-salted almonds of the nonpareil variety, supplied by the Almond Board of California (US grade extra no. 1). Participants were provided with snack information sheets and dietary advice from the research dietitian including instructions to only consume study snacks in between meals, maintain their habitual mealtime, eating habits and fruit consumption as well as avoid the consumption of additional nuts or nut products. To verify participants’ compliance with the dietary advice, telephone 24 h recalls and four-day estimated portion size food records were conducted.

Participants were provided with 20% EER from the muffins or almonds, which for a 2000 kcal EER, equated to five control muffins daily (400 kcal in (80 kcal/muffin)), or 63 g/d almonds (100 g almonds contains 634 kcal). Comparison of nutritional composition between almonds and control muffins is detailed in Appendix A. The ratio of sweet: savory muffins consumed was 70:30, with a mean of 20% of energy from free sugars in total muffins.

### 2.4. Anthropometry, Blood Pressure Measurements, and Blood Sampling

WC was measured using a measuring tape and body weight using electronic scales (Tanita BC-418MA, Tanita Ltd., Middlesex, UK). OMRON M2 Basic Intellisense monitors (OMRON Healthcare UK Ltd., Milton Keynes, UK) were used to measure seated blood pressure according to British Hypertension Society guidelines [33]. At screening, fasting blood was collected for analysis of glucose, lipids, liver function and full blood count. For baseline and endpoint visits, fasting blood samples were taken for analysis of plasma lipids, such as total, HDL-, and LDL-cholesterol and TAG and calculated TC:HDL-C ratio, as reported previously [28]

### 2.5. HRV Measurement

HRV assessments encompass acute fluctuations in IBIs (beat-to-beat HRV, parasympathetically regulated), as well as longer-phase oscillations (also reflecting other rhythms related to thermoregulation, hormonal fluctuation and circadian patterns). To measure real-time HRV in the supine position during mental stress, a small, light-weight, chest-worn wireless 2-lead ambulatory heart rate/ECG monitor (eMotion Faros 180, Mega Electronics Ltd., Kuopio, Finland) was fitted. Mental stress was induced by requiring the participant to complete the Stroop test—a test with colored words that has been used previously for this purpose during cardiovascular measurements [34,35]. The 5-min Stroop test was conducted 15 min after a 5-min resting HRV recording.

Cardiscope™ Analytics software (HASIBA Medical GmbH, Graz, Austria) was used to analyze the ECG data. Linear time-domain HRV parameters included the mean of the standard deviations of the normal-to-normal (NN) intervals (SDNN; indicating overall HRV) and root mean square of successive differences of NN intervals (RMSSD; indicating beat-to-beat, respiration-driven variability, and representing parasympathetic regulation) and linear frequency-domain HRV parameters included high-frequency (HF) power (reflects parasympathetic respiratory modulation) and LF/HF (previously assumed to reflect relative sympathetic to parasympathetic activity, but due to conflicting evidence for the role of the LF component in signifying sympathetic activity, this has been challenged) [9,36]. HF is expressed both as absolute unit (ms^2^) and normalized unit (nu), the latter being an adjustment for changes in total spectral power (except VLF). The short duration of stress test in our study precluded the inclusion of VLF power (0.003–0.04 Hz), a parameter of longer-phase oscillations in autonomic regulation and the renin-angiotensin system [9]. The non-linear parameter, the Poincaré ratio (SD1/SD2; the ratio of the SD of beat-to-beat IBI variability (SD1) against the SD of long-term IBI variability (SD2), indicating normality of sinoatrial firing patterns), is also measured as a non-linear measure analysis parameter [9].

### 2.6. Statistical Analysis

The sample size required per treatment arm was 50 subjects with 90% power and a two-tailed alpha set at 0.05 to detect significant between-treatment differences in flow-mediated dilation (FMD) of 1.25% unit difference (SD 1.9) [37], a primary outcome for the study. Allowing for potential drop-outs, the researchers enrolled 109 subjects to commence the run-in phase. HRV measures were secondary outcomes of the ATTIS study. Due to poor quality food diaries, a per protocol analysis was performed on nutrient intake data; other data were missing due to attrition, human error or technical failure. IBM SPSS 25 was used for statistical analysis. Normality of data was assessed visually using histogram and Q–Q plot of residuals. Baseline data are shown as mean value and standard deviation (SD), unless not normally distributed, and then, shown as median (IQR). Treatment effects are presented as the adjusted mean differences between groups at endpoint, with 95% CI. Chi-square test was conducted to investigate whether there were differences in sex and ethnicity between the control and almond groups at baseline. To examine whether there were differences in other baseline characteristics and also in baseline data between the two treatment groups, an independent t-test was used for normally distributed data and Mann–Whitney U test was used for non-normally distributed data. To investigate the significance of treatment effects between the two groups, analysis of covariance (ANCOVA) was used, adjusting for baseline value and baseline BMI. A two-sided *p*-value of <0.05 was considered to show statistical significance.

## 3. Results

### 3.1. Participant Characteristics

There were 109 participants recruited, 107 randomized and 105 completed the study. Drop-outs were due to non-compliance, time commitment issues and gastrointestinal intolerance of almonds. Table 1 shows characteristics at screening of participants who were randomized to treatment. Of 107 participants randomized, 75 were females and 32 were males, and the average age was 56 y. The CONSORT diagram in Appendix A shows the full stream of participant enrolment, allocation to treatment and disposition.

### 3.2. Nutrient Intakes and Blood Markers

Total energy intake of both treatment groups was not different, as shown in Table 2. The almond group had significantly higher intakes of dietary fiber, fat, mono- and polyunsaturated fatty acids, the ratio of unsaturated to saturated fatty acids, potassium, magnesium, vitamin E and riboflavin. Intake of total carbohydrate, starch, free sugar and sodium was shown to be significantly lower in the almond group.

There were no significant differences in changes in body composition (Appendix A), nor markers of insulin sensitivity (data reported elsewhere [28]). Plasma non-HDL and LDL were significantly lowered by 0.22 mmol/L (95% CI −0.42, −0.01) and 0.25 mmol/L (95% CI −0.45, −0.04) respectively following almonds relative to control, as previously reported [28].

### 3.3. Heart Rate Variability at Rest and during Mental Stress

Compared to resting values, mean 5-min NN decreased during mental stress in both treatment groups (1030.3 ± 156.4 vs. 921.1 ± 148.6, paired t-test *p* ≤ 0.001). As shown in Table 3, in the resting state, there were no significant differences between treatment groups in the change in HRV indices following intervention. However, during mental stress, HF power was higher following almond treatment by 124 ms2 (95% CI 11, 237), relative to control. Moreover, LF/HF was lower by −1.0 (95% CI −1.9, −0.1) relative to control. No differences were found in other indices during mental stress.

## 4. Discussion

Mental stress can lower HRV, which is associated with increased risk of developing CVD [7,8]. Improvements in diet quality have the potential to increase HRV, but evidence for a direct causal relationship is limited. We report the novel finding that snacking on whole almonds for six weeks, compared with isocaloric snacks with a more typical nutrient profile (high in saturated fats, starch and free sugars and low in dietary fiber), increased HRV parameters of parasympathetic activity during acute mental stress. This could be due to lower levels of background daily stress in the almond group, improvements in neurological autonomic function, or an increase in cardiac tissue responsivity to ANS neurotransmission and/or hormonal modulation. Although the underlying mechanism of effect is unclear, the results of the study suggest that a simple dietary modification resulting in increased intake of micronutrients, dietary fiber and unsaturated fatty acids and reduced intake of free sugars and sodium improved vagal tone during mental stress.

Variability in beat-to-beat intervals, assessed by HF power in the frequency domain, is associated with respiration, known as respiratory sinus arrhythmia (RSA). RSA is mediated by the parasympathetic nervous system modulated by vagal motor neurons linked with the lung inflation reflex [38,39,40]. Only a very limited amount of research has been done previously on the effects of tree nuts on HRV. In agreement with our findings, higher HF power was previously observed by Sauder et al. during two acute stress tests, i.e., mental arithmetic and hand cold pressor, following 4-week pistachio nut consumption at 20% of energy intake, as well as increased rMSSD and LF power [41]. In our study, there were no treatment effects observed for rMSSD and LF, possibly due to variability across studies in baseline stress levels and methodological differences such as type of mental stressor and measurements being made in the seated position in contrast to our study where measurements were made in the supine position. In the current study, the ratio of LF to HF power was shown to be decreased in almond group, which could suggest that replacing typical snacks with almonds might tip the balance of sympathetic to parasympathetic nervous system activity to a more favorable one. However, the interpretation of LF/HF ratio is controversial; the original belief that LF power is related to sympathetic modulation has been widely discounted as results of experiments inducing pharmacological sympathetic blockade and other manipulations of sympathetic activity [42]. Thus, the difference between treatments in LF/HF ratio is likely to be largely attributable to increased vagal tone rather than any reduction in sympathetic outflow [43,44].

Although weight loss is known to increase parasympathetic activity [45], there were no differences between groups in the change from baseline in body weight and adiposity, nor energy intake [28]. The established effects of almond consumption in lowering plasma low-density lipoprotein (LDL) cholesterol concentrations may have some bearing on HRV responses, as the literature reports that plasma TC and LDL are inversely associated with HR and HRV [46,47]. Statin treatment was associated with improved HRV in 40 hypercholesterolemic patients with or without CAD [48] and healthy individuals with 48 h sleep deprivation [49]. Hypercholesterolemia elevates reactive oxygen species (ROS) and oxidative stress in vessel walls and induces inflammation, causing dysfunctional nitric oxide synthase (eNOS) activity, an enzyme catalyzing nitric oxide (NO) production and greater degradation of NO [50], resulting in dysregulation of vascular tone [51]. Baroreceptors sense systemic arterial pressures via stretching, and impaired vascular tone is likely to disrupt baroreceptor sensitivity, resulting in reduced baroreflex control of HR, potentially attenuating HRV [52]. Therefore, reductions in LDL cholesterol concentrations following almond intervention may have indirectly contributed to enhancement of baroreceptor sensitivity and maintenance of vagal tone. This proposed cardiovascular mechanism is strengthened by our finding reported previously that NO-mediated vasodilation was increased following the almond intervention in the same study [28].

Lower dietary glycemic load as a result of displacing typical snack nutrient intakes with almonds may be an important factor in the improved HRV observed under mental stress, as previous studies have observed increased LF/HF and reductions in total power from spectral analysis following oral glucose loads [53,54,55,56]. Insulin secretion is implicated in this effect, since hyperinsulinemia can reduce the functioning of the sinoatrial node and alter ANS activity [53,54]. Furthermore, the literature also demonstrated that glucose intake and hyperinsulinemia are dose-dependently associated with the level of circulating NE, a sympathetic-induced neurotransmitter [57]. Sodium and potassium levels may also play a part. Potassium is inversely associated with aldosterone, a hormone regulating sodium reabsorption which is involved in the renin-angiotensin-aldosterone system (RAAS). Lower sodium leads to reduced water retention in the kidney, smaller blood volume and lower blood pressure [58]. Blood pressure maintenance could promote baroreflex sensitivity and influence sympathetic activity [52].

Almonds are a good source of magnesium and in this study, the almond group had higher magnesium intakes relative to the control group. Animal experiments demonstrated that magnesium has anti-arrhythmic effects involving improvement in sinus rhythm [59], but human data are inconsistent in treatment of arrhythmia [60,61,62] and in associations with HRV [63,64,65]. Almonds are rich in vitamin E, and 4-month supplementation in a double-blind RCT improved HRV by increasing RR interval, TP and HF and reducing LF and LF:HF in T2D patients, possibly due to a reduction in oxidative stress [66]. Although it is not possible to pinpoint the components of almonds that may be responsible for the greater cardiac resilience to mental stress observed in the current study, it is possible that the relative augmentation in vagal tone during the Stroop test by almond consumption was a combined result of all or many of the mechanisms discussed.

The main strength of our study was the randomized, controlled, single-blinded study design involving control test snacks that represented typical snacks consumed in the adult population. Almond consumption did not affect either body weight, central adiposity (waist circumference) or overall adiposity (body fat percentage), and therefore, fat mass loss was not a confounding factor [67,68,69,70]. Although we recruited healthy adults at risk of developing CVD, recording of self-reported mental and mood conditions before the mental stress task and HRV measurements were not included and may be a limitation. The Stroop test was conducted following 10 min recovery after blood-pressure cuff inflation applied within a flow-mediated dilation (FMD) measurement that could cause physical stress, which might affect baroreceptor-derived autonomic outflow, and therefore, influence the resting measurements. Furthermore, it would be of interest to determine the response in recovery HRV after the Stroop test. Recruiting patients with obesity or type-2 diabetes, determinants of impaired sympathovagal balance, should be a priority in future research to understand more the effects of whole almonds on HRV in at risk populations.

## 5. Conclusions

Snacking on whole almonds in place of typical snacks can increase HRV parameters of parasympathetic activity in response to mental stress, indicating improved cardiac autonomic function. Incorporating tree nuts as daily snacks is encouraged as a positive lifestyle change and enhances cardiovascular health, not only by lowering cholesterol, but also potentially by increasing resilience to mental stress.

## Figures and Tables

**Table 1 nutrients-12-01828-t001:** Subject characteristics at screening for those randomized to treatment (mean ± SD).

	Control, *n* = 51	Almond, *n* = 56
Age, y	56.0 ± 10.7	56.3 ± 10.3
Sex, M/F, *n*	15/36	17/39
Ethnicity (Black/South Asian, Southeast Asian and Middle Eastern/Far East/White/Other), *n*	2/6/2/34/7	9/7/3/34/3
Cardiometabolic score	4.2 ± 2.1	4.5 ± 2.0
BMI, kg/m^2^	26.7 ± 4.5	27.3 ± 4.4
WC, cm	93.3 ± 12.5	93.6 ± 12.5
% body fat	32.7 ± 8.5	34.4 ± 8.4
cSBP, mmHg	124.4 ± 15.1	126.2 ± 17.6
cDBP, mmHg	80.6 ± 7.7	83.8 ± 10.8
Glucose, mmol/L	5.1 ± 0.6	5.1 ± 0.5
TC, mmol/L	5.6 ± 1.2	5.6 ± 1.0
TAG, mmol/L	1.2 ± 0.5	1.2 ± 0.6
LDL-C, mmol/L	3.5 ± 1.0	3.4 ± 0.9
HDL-C, mmol/L	1.6 ± 0.5	1.6 ± 0.5
TC:HDL	3.6 ± 0.9	3.7 ± 1.1

Ethnicity was determined by self-reporting. Blood measures taken in a fasting state. BMI, body mass index; WC, waist circumference; cSBP, clinic systolic blood pressure; cDBP, clinic diastolic blood pressure; TC, total cholesterol; TAG, triacylglycerol; LDL-C, low-density cholesterol; HDL-C, high-density cholesterol; TC:HDL, total cholesterol:high-density cholesterol.

**Table 2 nutrients-12-01828-t002:** Nutrient intakes estimated from 4-d food diaries at baseline (prior to run-in) and the final week of the dietary intervention.

	Control, *n*_max_ = 40 ^2^		Almond, *n*_max_ = 40 ^2^		Mean Comparison between Groups	*p*-Value
	Baseline	Change	Baseline	Change		
Energy intake ^1^, kcal/d	2088.9 ± 538.5	−5.8 (−124.7, 113.2)	1769.4 ± 475.0	−85.3 (−204.3, 33.7)	−79.5 (−251.8, 92.8)	0.361
Protein, %E	15.4 ± 3.8	0.5 (−0.5, 1.5)	15.9 ± 3.6	1.0 (0.0, 2.0)	0.5 (−0.9, 1.9)	0.466
Carbohydrate, %E	43.3 ± 7.1	1.7 (−0.1, 3.5)	41.8 ± 6.6	−7.6 (−9.4, −5.8)	−9.3 (−11.9, −6.8) *	<0.001
Starch, %E	23.9 ± 5.1	2.5 (0.9, 4.1)	23.5 ± 5.3	−4.5 (−6.1, −2.9)	−7.0 (−9.3, −4.8) *	<0.001
Free sugars, %E	5.9 ± 3.8	0.4 (−0.5, 1.2)	5.5 ± 2.8	−2.6 (−3.5, −1.8)	−3.0 (−4.2, −1.8) *	<0.001
Dietary fibre ^1^, g/d	23.8 ± 6.2	−1.9 (−4.5, 0.6)	20.7 ± 7.7	5.5 (3.0, 8.1)	7.4 (3.8, 11.1) *	<0.001
Fat, %E	36.5 ± 6.5	−2.6 (−4.3, −0.8)	37.1 ± 6.2	8.3 (6.5, 10.0)	10.8 (8.4, 13.3) *	<0.001
SFA, %E	12.3 ± 3.6	−0.6 (−1.3, 0.1)	12.5 ± 3.7	−1.4 (−2.1, −0.6)	−0.7 (−1.8, 0.3)	0.153
MUFA, %E	11.5 ± 3.4	−1.1 (−2.4, 0.0)	12.4 ± 3.7	8.6 (7.4, 9.8)	9.8 (8.1, 11.5) *	<0.001
PUFA, %E	5.9 ± 2.5	−0.8 (−1.4, −0.1)	5.9 ± 1.7	2.0 (1.4, 2.6)	2.8 (1.9, 3.7) *	<0.001
Unsaturated: saturated fatty acid ratio	1.5 ± 0.5	−0.1 (−0.3, 0.1)	1.6 ± 0.7	1.1 (0.9, 1.3)	1.3 (1.0, 1.5) *	<0.001
Sodium, mg	2151.2 ± 766.3	179.7 (−15.8, 375.3)	1926.1 ± 866.1	−490.8 (−686.4, −295.3)	−670.6 (−948.6, −392.6) *	<0.001
Potassium ^1^, mg	3028.9 ± 936.2	−352.5 (−590.4, −114.5)	2534.7 ± 854.5	221.3 (−16.7, 459.3)	573.8 (231.0, 916.6) *	0.001
Calcium ^1^, mg	868.4 ± 455.8	24.2 (−57.6, 106.0)	703.8 ± 242.5	57.3 (−24.5, 139.0)	33.1 (−84.0, 150.2)	0.575
Magnesium ^1^, mg	368.7 ± 180.9	−36.0 (−68.9, −3.0)	278.5 ± 92.1	112.6 (79.7, 145.5)	148.6 (100.9, 196.3) *	<0.001
Vitamin E, mg	10.7 ± 3.7	−1.9 (−3.5, −0.4)	8.9 ± 4.0	13.5 (11.9, 15.0)	15.4 (13.2, 17.6) *	<0.001
Riboflavin, mg	1.8 ± 1.6	−0.1 (−0.3, 0.0)	1.5 ± 0.8	0.4 (0.2, 0.6)	0.5 (0.3, 0.8)	<0.001
Niacin, mg	16.1 ± 8.9	−1.6 (−3.3, 0.1)	14.7 ± 9.4	−0.4 (−2.1, 1.3)	1.2 (−1.3, 3.6)	0.339

Values of change and main comparison between groups are presented as mean (95% CI). ANCOVA was used, adjusted for baseline value and baseline BMI. * *p* < 0.05 indicating a significant difference. ^1^ Baseline value was different between control and almond group. ^2^ Data were analyzed using 40 diaries collected from each group. Missing data are due to poor quality diet diaries or failure to complete by participant.

**Table 3 nutrients-12-01828-t003:** Heart rate variability values measured during 5-min periods of rest and mental stress (Stroop test), following randomization to almond and control snacks.

	Control, *n*_max_ = 51 ^1^		Almond, *n*_max_ = 54 ^1^		Main Comparison between Groups at Endpoint ^4^	*p*-Value between Groups at Endpoint
	Baseline ^2^	Endpoint	Baseline ^2^	Endpoint		
*Resting*						
NN, ms	1009 ± 166	1006 (977, 1035)	1050 ± 146	1012 (984, 1040)	6 (−34, 47)	0.760
HR	60.1 ± 10.7	60.5 (58.7, 62.3)	58.0 ± 7.6	61.0 (59.3, 62.7)	0.5 (−2.0, 3.0)	0.704
rMSSD, ms	32.6 (24.6)	37.7 (34.0, 41.5)	31.9 (25.7)	37.2 (33.5, 40.8)	−0.6 (−5.8, 4.7)	0.831
SDNN, ms	54 ± 27	49 (44, 544)	47 ± 21	50 (45, 55)	1 (−6, 8)	0.740
SD1/SD2	0.5 ± 0.3	0.4 (0.4, 0.5)	0.4 ± 0.1	0.4 (0.4, 0.5)	0 (−0.1, 0)	0.651
HF, ms ^2^	373 (631)	599 (468, 730)	322 (541)	514 (391, 636)	−85 (−265, 95)	0.348
HFnu	0.40 ± 0.17	0.41 (0.36, 0.46)	0.41 ± 0.14	0.42 (0.37, 0.46)	0.01 (−0.06, 0.07)	0.871
LF:HF	1.5 (1.7)	1.9 (1.5, 2.2)	1.4 (1.3)	1.9 (1.5, 2.2)	0 (−0.5, 0.5)	0.924
*Mental stress (Stroop test)*						
NN ^3^, ms	888 ± 166	935 (907, 963)	950 ± 127	924 (899, 950)	−11 (−49, 28)	0.310
HR^3^	68 ± 10	65.4 (63.5, 67.3)	64 ± 8	66.2 (64.5, 67.9)	0.8 (−1.8, 3.4)	0.543
rMSSD, ms	38.2 ± 22.4	31.8 (27.1, 36.5)	33.8 ± 14.0	34.0 (29.8, 38.2)	2.2 (−4.1, 8.5)	0.494
SDNN, ms	54 (32)	44 (41, 47)	45 (22)	48 (44, 51)	4 (−1, 8)	0.137
SD1/SD2	0.4 (0.2)	0.4 (0.3, 0.4)	0.4 (0.1)	0.4 (0.4, 0.4)	0 (0, 0.1)	0.420
HF, ms ^2^	394 (473)	281 (197, 364)	264 (495)	405 (331, 480)	124 (11, 237)	0.031 *
HFnu	0.31 ± 0.10	0.26 (0.22, 0.30)	0.32 ± 0.12	0.32 (0.28, 0.35)	0.06 (0, 0.11)	0.040 *
LF:HF	2.3 (1.5)	3.6 (3.0, 4.3)	2.0 (1.7)	2.6 (2.0, 3.2)	−1.0 (−1.9, −0.1)	0.023 *

Endpoint values and main comparison between groups at endpoint are presented as mean (95% CI). ^1^ Not all data were analyzed due to technical problems. Resting NN, HR, rMSSD, SDNN and SD1/SD2: *n* = 40 (control) and 43 (almond). Resting HF, HFnu, and LF:HF: *n* = 35 (control) and 40 (almond). Mental stress (Stroop test) NN and HR: *n* = 35 (control) and 44 (almond). Mental stress (Stroop test) rMSSD, SDNN and SD1/SD2: *n* = 36 (control) and 44 (almond). Mental stress (Stroop test) HF, HFnu and LF:HF: *n* = 28 (control) and 35 (almond). ^2^ Mean ± SD for baseline data that are normally distributed. Median (IQR) for other data as they are non-normally distributed. ^3^ Baseline value was different between control and almond group; independent *t*-test was used for normally distributed data while Mann–Whitney U test was used for non-normally distributed data; *p* < 0.05 indicating a significant difference. ^4^ ANCOVA, adjusted for baseline outcome value and baseline BMI (mean difference almonds—control at endpoint). * *p <* 0.05. NN, normal-to-normal intervals; HR, heart rate; rMSSD, root mean square of successive R-R interval differences; SDNN, standard deviation of normal-to-normal (NN) intervals; HF, absolute power of the high-frequency band (0.15–0.04 Hz); HFnu, normalized HF (HFnu = HF/(HF + LF)).

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
