# Peer review of "Snacking on Whole Almonds for Six Weeks Increases Heart Rate Variability during Mental Stress in Healthy Adults: A Randomized Controlled Trial"

_nutrients, 2020, doi:10.3390/nu12061828_

Round 1

Reviewer 1 Report

In the current study by Vita Dikariyanto etal, authors studied the impact of almonds on heart rate variability (HRV). They found that snacking almonds could enhance HRV related to parasympathetic stimulation during mental stress and hence improve cardiac autonomic function. The study has moderate patient group with good experimental design and execution. However, some minor concerns are as below.

  • The tables should be labelled properly (table 3 is labelled as table 1).
  • The authors should provide a separate table for all the abbreviations used in the manuscript.
  • The authors should compile a separate section for the strengths and limitations of the study.

Reviewer 2 Report

The paper describes a secondary analysis of the ATTIS trial with the aim of showing that snacking on almonds may ameliorate HRV parameters with respect to control snacks. In the following, some comments referring to the statistical analyses are reported.

2.2. Study design:

a) Please report whether or not participants, those administering the interventions, and those assessing the outcomes were blinded to group assignment. If done, how the success of blinding was evaluated.

b) In the description of the randomization schema, please report the value of the bias probability that was used in the allocation to minimize the overall imbalance among groups.

2.6. Statistical analysis

a) Please, report the sample size that results from the power analysis; with a difference between the two groups equal to 1.25 and assuming a common standard deviation in the two populations equal to 1.9, the sample size required to achieve a 90% of power with a two sided significance level of 5%, is 50 subjects per treatment arms. Why the authors enroll 109 subjects? If they consider an increment in the sample size to account for possible drop-out, this should be explicitly reported.

b) Due to the randomized nature of the study, I would suggest to remove the statistical comparison of baseline characteristics between the two groups as any significance would be, by definition, a Type I error.

c) Please report explicitly that a per-protocol analysis was performed and explain the reasons for such approach.

3.1. Participant characteristics

a) Table 1. Please refers to the comment b) previously done.

3.3. Heart rate variability at rest and during mental stress

b) The results reported in table " Heart rate variability values measured during mental stress" require several amendments.

            b1) The authors seem to adopt two different notations to report the IQR. In some cases the IQR seems to be presented as difference between the first and the third quartile while in others the authors report explicitly the two quartiles. Please, make the notation uniform.

            b2) It is unclear why the authors use different descriptions for the same variable in the two time points (baseline and endpoint). For instance, the distribution of the variable NN, ms is described by using the mean ± standard deviation at baseline and the median with IQR at the end of follow up. This makes any within-group difference very difficult to catch. See also comment below.

            b3) The authors always have used a parametric approach (ANCOVA) to adjust between group comparisons for baseline outcome value and BMI, even for those variables that, according to the notation used by the same authors, presented a consistent asymmetry. Although the ANCOVA model is based on the change-from-baseline values, did the authors verify that these values could be reasonably analyzed using such an approach?. If yes, maybe the authors could consider to report for each variable, in both groups, the mean ± standard deviation of the change-from-baseline values instead of the two time points. This could also remove the concern expressed in the previous point. Otherwise, the authors should use different multivariable approach to compare the two groups, namely the median regression.

Minor remarks

In table 2, the 7 superscript should be replaced by an asterisk.

In table "Heart rate variability values measured during mental stress (Stroop test), following randomization to almond and control snacks." the superscripts should be rearranged as they not respect the information provided in the table legend.

Round 2

Reviewer 2 Report

The authors properly addressed all my previous concerns.